# FEM Simulation of Frequency-Selective Surface Based on Thermoelectric Bi-Sb Thin Films for THz Detection

**Anastasiia Tukmakova** [1,*] **, Ivan Tkhorzhevskiy** [1] **, Artyom Sedinin** [1] **, Aleksei Asach** [1] **, Anna Novotelnova** [1] **, Natallya Kablukova** [1] **, Petr Demchenko** [2] **, Anton Zaitsev** [2] **, Dmitry Zykov** [2] **and Mikhail Khodzitsky** [2]

1    Faculty of Energy and Ecotechnology, ITMO University, 197101 St. Petersburg, Russia;
     iltkhorzhevskiy@itmo.ru (I.T.); adsedinin@itmo.ru (A.S.); avasach@itmo.ru (A.A.);
     novotelnova@itmo.ru (A.N.); n.kablukova@itmo.ru (N.K.)
2    Terahertz Biomedicine Laboratory, ITMO University, 197101 St. Petersburg, Russia; 192249@niuitmo.ru (P.D.);
     a.zaytsev@niuitmo.ru (A.Z.); dvzykov@itmo.ru (D.Z.); khodzitskiy@yandex.ru (M.K.)
*    Correspondence: astukmakova@itmo.ru

**Abstract:** Terahertz (THz) filters and detectors can find a wide application in such fields as: sensing, imaging, security systems, medicine, wireless connection, and detection of substances. Thermoelectric materials are promising basis for THz detectors' development due to their sensitivity to the THz radiation, possibility to be heated under the THz radiation and produce voltage due to Seebeck effect. Thermoelectric thin films of Bi-Sb solid solutions are semimetals/semiconductors with the band gap comparable with THz energy and with high thermoelectric conversion efficiency at room temperature. Detecting film surface can be transformed into a periodic frequency selective surface (FSS) that can operate as a frequency filter and increases the absorption of THz radiation. We report for the first time about the simulation of THz detector based on thermoelectric Bi-Sb thin-filmed frequency-selective surface. We show that such structure can be both detector and frequency filter. Moreover, it was shown that FSS design increases not only a heating due to absorption but a temperature gradient in Bi-Sb film by two orders of magnitude in comparison with continuous films. Local temperature gradients can reach the values of the order of $100 \, \mathrm{K \cdot mm^{-1}}$. That opens new perspectives for thin-filmed thermoelectric detectors' efficiency increase. Temperature difference formed due to THz radiation absorption can reach values on the order of 1 degree. Frequency-transient calculations show the power dependence of film temperature on time with characteristic saturation at times around several ms. That points to the perspective of reaching fast response times on such structures.

**Keywords:** THz frequency range; thermoelectric; FEM simulation; sensor; detector; filter; frequency-selective surface; electromagnetic heating

## 1. Introduction

According to many studies [1], as well as already implemented applications, the terahertz (THz) frequency range is extremely promising for modern THz science and technology. Many substances like water absorb terahertz radiation which is nonionizing due to the low quantum energy of such radiation. These features make it possible to use THz radiation in the study of biological tissues and many organic substances [2]. Many materials selectively absorb radiation in the THz frequency range, which makes it possible to use it in various engineering systems [3], for example, for diagnostics [4] and safety purposes [5]. Furthermore, such applications like imaging [6], wireless communications [7], and spectroscopy [8] can be carried out using THz devices.

Currently, there is a need for THz radiation detectors that would simultaneously have a high response rate and sensitivity and would also work at room temperature without additional cooling. The solution to the problem may be based on photothermoelectric or thermoelectric effects [9]. Photothermoelectric effect consists in the heating of the material

due to radiation absorption and generation of electric voltage in a gradient temperature field due to Seebeck effect. Thermoelectric detectors can be a good alternative to bolometers, photoconductive antennas, and Golay cells [10,11]. Thin films of graphene, black phosphorus, transition metal dichalcogenides, nitrides, and carbonitrides are already widely studied as photothermoelectric detectors at room temperature [12]. Thermoelectrics can be used for detection from visible to infrared spectrum due to the possibility of precious temperature gradient control at the nanoscale volumes in resonant structures [13]. Black-phosphorus [14] and Se-doped black-phosphorus [15] nanotransistors have been proposed as THz detectors working on thermoelectric response. Thermoelectric graphene-based detector has been presented in [16] and in [17] thermoelectric effect has been used to increase the responsivity of graphene-based THz detector. In [18] Au-based thermoelectric antenna has been fabricated for 0.6 THz frequency detection. To increase a temperature gradient in the system a hot junction width has been reduced to 100 nm. FSS and metasurfaces also have been studied as absorption enhancers for thermoelectric energy harvesters in infrared range [19,20].

For the fabrication of THz detector based on thermoelectric effect the material of detecting surface must possesses two main characteristics: (i) suitable optical properties allowing the material response to THz radiation, such as permittivity, absorption, conductivity and (ii) high thermoelectric efficiency. At the same time, heating due to radiation absorption itself is not so important as temperature gradient. Material must be in a sufficient temperature gradient field to produce higher voltage. High temperature gradient can be achieved due to system geometry variation (narrow or thin regions, cross-sections reduction, corners, etc.).

Thermoelectric efficiency of energy conversion depends on material figure of merit ZT [21]. The materials based on Bi, Sb, Te, and Se have one of the highest figures of merit at room temperature among other known thermoelectrics [22], for example, Bi-Sb-Te p-type alloys with different doping can achieve ZT = 1.15–1.36, and n-type Bi-Te-Se alloys can achieve ZT = 0.7–0.8, both at 300 K. Thin and nanostructures based on these materials are promising for thermoelectricity [23] due to the the reduced thermal conductivity because of the phonon scattering. Thin structures are also compact which is a good aspect for microsystems, sensing, and imaging.

These materials possess a narrow bandgap comparable with an energy of THz photons that can be used for practical detection applications. It was shown in [13] that bismuth telluride and antimony telluride nanostructures can be a perspective basement for nanophotonic detectors with high responsivity and temporal response (337 μs). In [24] a Bi/Bi-Sb-based thermoelectric THz antenna showed a temporal response of 22 μs without additional cooling. We showed in previously published works [25] that Bi-Sb thin films prepared by vacuum thermal deposition have perspective optical properties in the range from 0.2 to 0.8 THz: subpicosends relaxation time of carriers, high values of permittivity, and conductivity. The composition with the antimony content of 12 percent ($Bi_{88}Sb_{12}$) seems to be the most promising. This composition has the highest response to THz radiation switching on: voltage drop of 0.1 mV along the $Bi_{88}Sb_{12}$ sample in comparison with 0.01 mV for $Bi_{97}Sb_3$ and $Bi_{92}Sb_8$ samples [26]. Temperature difference of several degrees in $Bi_{88}Sb_{12}$ film has been shown. One more advantage of the studied Bi-Sb films is that vacuum thermal deposition is not time and money consuming and it is a relatively easy method of films fabrication.

Further improvement of detection surface can be implemented with the frequency selective surface fabrication. Frequency-selective surfaces (FSS) are two-dimensional structures consisting of periodic unit cells with a specific configuration that determines the interaction with radiation of a certain set of frequencies. Depending on the geometry [27–29] of the unit cell and the materials used, FSS selectively transmits, reflects, and absorbs the electromagnetic radiation. Their response to radiation also depends on its angle of incidence and polarisation. The unit cells of FSS are mainly represented by split ring resonators [30],

wires [31], chiral and cross-like elements [32,33], holes or patches of different configuration, etc [34].

Among plenty of different applications [35] of FSS in absorbers [36], radomes [37], waveguides, resonators [38], microwave ovens, and communications [39], there is such radiation filtering. Due to high transmission or reflection coefficient, FSS are a great solution for bandpass or bandstop (notch) filters. Generally, these structures usually consist of a metal mask on a nonmetallic substrate. Due to fabrication capabilities, they may operate at very high frequencies [40].

For special applications like sensing or detection, metal in filtering structures may be replaced by a semiconductor or semimetal. In our case, we propose to use $Bi_{88}Sb_{12}$ layer with periodic cross-shaped elements. In this case, detecting surface can also be used as a frequency filter. That will increase the responsivity of a detector due to the increased absorption. In addition, changed surface geometry with periodic structure can affect the temperature field, can cause local increase of temperature gradient, and results in the higher voltage generation. Few works about FSS based on thermoelectric films are presented in literature. For example, Thermally tunable FSS based on $Ba_xSr_{1-x}TiO_3$ thermoelectric film has been proposed as a filter with transmission maximum at 0.8–0.9 THz [41]; change of FSS temperature resulted in the resonance shift.

The combination of frequency filter and detector in one device, implemented through the FSS fabrication is perspective and complex challenge. Practical manufacturing of thin-film-based FSS is a challenging task. Numerical modelling is widely used in photonics for detecting devices development. Prior numerical calculations are required for the simulation of a device with finite size and geometry, taking into the consideration electromagnetic and thermal phenomena, structure periodicity, and boundary condition close to real operation conditions. Current work aims to perform preliminary calculations for further thermoelectric FSS fabrication for THz radiation filtering and detection.

This paper reports the finite elements simulation of FSS based on Bi-Sb thin film on the top of a dielectric substrate. Several questions were raised:

- the possibility to use Bi-Sb films on dielectric substrates as FSS with satisfactory quality factor;
- the possibility to use Bi-Sb-based FSS for THz detection due to electromagnetic heating;
- the determination the ability of FSS geometry to increase a temperature gradient in the Bi-Sb film.

## 2. Materials and Methods

### 2.1. Geometry of a Frequency-Selective Surface

The simulated object is the periodic element of a frequency-selective surface (Figure 1a): film with the cross-shaped element cut from its volume and deposited on the substrate. In current work such surface works as a frequency-selective mirror. The interaction of electromagnetic wave with this surface can be characterised by a frequency-dependent reflection amplitude spectrum (Figure 1b). This spectrum has a resonance at the specific frequency and reflection stopband (transmission passband) is formed at a specific frequency range. The smaller the bandwidth ($\Delta f$, Figure 1b), the higher the quality factor $Q$ of filter. Quality factor (Q-factor) can be found from the following formula:

$$Q = f_r / \Delta f. \tag{1}$$

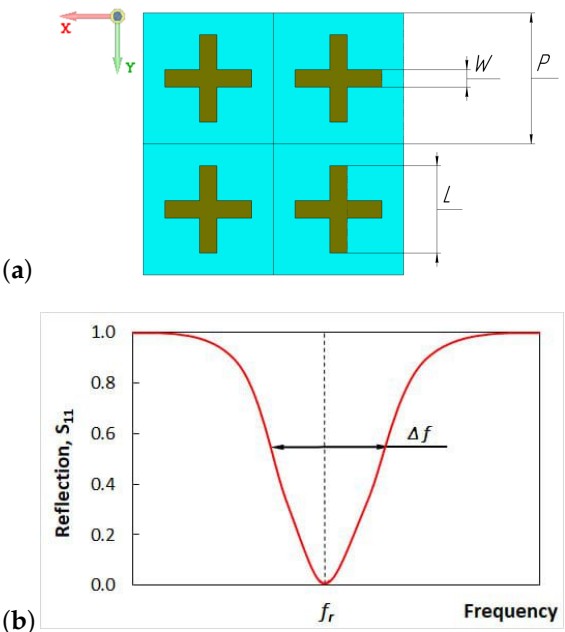

(a)

(b)

**Figure 1.** (**a**) Geometric parameters of a periodic frequency-selective surface: *P* is a structure period, *L* is a cross length, and *W* is a cross width. (**b**) Typical view of reflection amplitude spectrum: $f_r$ is a resonance frequency, Δf is a bandwidth, that can be found as a bandwidth corresponding to the semiamplitude.

Change in the geometric parameters of the FSS allows the adjustment of the reflection amplitude spectrum [42]. The change of period *P*, cross width *W*, and length *L* changes the position and Q-factor of the resonance.

General and side views of the simulated object are presented in Figure 2.

(a)

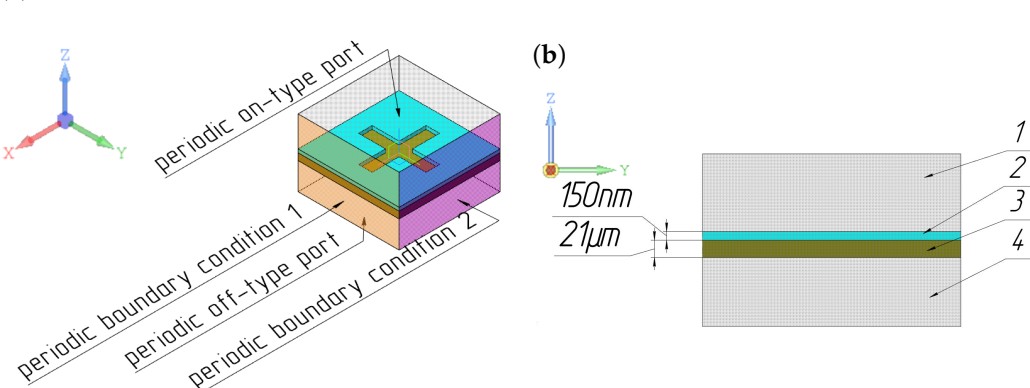

(b)

**Figure 2.** The geometry of the system used in the simulation (**a**) general 3-D view (**b**) side view. 1 and 4—vacuum (the height was a half of the wavelength), 2—Bi-Sb film, 3—dielectric substrate.

### 2.2. Materials

Thermoelectric film in simulation model represents the $Bi_{88}Sb_{12}$ of 150 nm thickness film obtained by the thermal vacuum deposition method [43] and having promising optical and thermoelectric properties [25]. Substrate material corresponded to mica and polyimide (PM). The properties used for the simulation are listed in Tables 1–4. Materials' properties have been taken at 0.1 and 0.14 THz as these frequencies are often used in commercially available equipment, e.g., Terasense Group Inc. [44]. The material properties have been obtained in [25] by THz time-domain spectroscopy. Briefly, the waveforms of THz pulses transmitted through the air, the substrate, and the film-on-substrate structure were recorded.

Using Fourier transformation, they were converted into complex electric field amplitude spectra which were used to calculate the substrate (mica or polyimide) permittivity [45] and, after that, the conductivity [46] and permittivity [47] of bismuth-antimony thin films.

**Table 1.** Relative permittivity, real part.

| Frequency, THz | Bi$_{88}$Sb$_{12}$ (On Mica) | Bi$_{88}$Sb$_{12}$ (on PM) | Mica | Polyimide |
|---|---|---|---|---|
| 0.1 | −20,000 | −18,000 | 5.2 | 2.5 |
| 0.14 | −22,000 | −16,000 | 5.2 | 2.5 |

**Table 2.** Relative permittivity, imaginary part.

| Frequency, THz | Bi$_{88}$Sb$_{12}$ (On Mica) | Bi$_{88}$Sb$_{12}$(on PM) | Mica | Polyimide |
|---|---|---|---|---|
| 0.1 | 127,000 | 45,000 | 0.2 | 0.9 |
| 0.14 | 104,000 | 38,000 | 0.2 | 0.9 |

**Table 3.** Electrical conductivity, real part, S/m.

| Frequency, THz | Bi$_{88}$Sb$_{12}$(On Mica) | Bi$_{88}$Sb$_{12}$(on PM) | Mica | Polyimide |
|---|---|---|---|---|
| 0.1 | 934,000 | 313,000 | $64 \cdot 10^{-5}$ | $1 \cdot 10^{-12}$ |
| 0.14 | 866,000 | 320,000 | $8 \cdot 10^{-4}$ | $1 \cdot 10^{-12}$ |

**Table 4.** Thermal conductivity, W/m·K.

| Component | Bi$_{88}$Sb$_{12}$ | Mica | Polyimide |
|---|---|---|---|
| cross-plane | 4 | 0.5 | 0.3 |
| in-plane | 8 | 5 | 0.3 |

*2.3. General Equations*

In current work, we perform the numerical simulation in Comsol Multiphysics. The simulation has been carried out with the use of "electromagnetic waves" and "heat transfer in solids" physics interfaces. The model was 3D and had been solved in frequency-transient solver. The model depending variables were electric field and temperature field. The work consisted of two general steps. Firstly, we solved the electromagnetic problem in order to optimise the filter geometry for specific frequencies (0.1 and 0.14 THz). The results of these calculations were the amplitude reflection spectra that allow the estimation of optimal system size and geometry. Secondly, we solved the adjacent task of electromagnetic heating of FSS. The results of these calculations were temporal dependencies of maximum temperature (or temperature difference) in the film.

The system of differential equations used for the simulation was the following:

Wave equation:

$$\nabla \times \hat{\mu}_r^{-1}(\nabla \times \mathbf{E}) - k_0^2(\hat{\epsilon}_r - i\hat{\sigma}/(\omega\hat{\epsilon}_0))\mathbf{E} = 0, \tag{2}$$

where $\mu_r$ is the complex relative permeability; $\mathbf{E}$ is the electric field; $k_0$ is the wave number in free space; $\epsilon_r$ is complex relative permittivity and $\epsilon_0$ is free space permittivity, respectively; $\sigma$ is the complex electrical conductivity; and $\omega$ is the angular frequency.

For the electric field displacement description, the dielectric loss model was used:

$$\hat{\epsilon}_r = \epsilon' - i\epsilon'', \tag{3}$$

where $\epsilon'$ and $\epsilon''$ are real and imaginary parts of permittivity.

Heat balance equation with electromagnetic heat source:

$$\rho c_p \frac{\partial T}{\partial t} - \nabla \cdot (\kappa \nabla T) = Q_e, \tag{4}$$

where $Q_e$ is the electromagnetic heat source, $\rho$ is density, $c_p$ is heat capacity, $T$ is the temperature, $\kappa$ is the thermal conductivity.

Electromagnetic heat source or electromagnetic loss density includes resistive and magnetic losses:

$$Q_e = Q_{rh} + Q_{ml} = \frac{1}{2} Re(\mathbf{J} \cdot \mathbf{E}^*) + \frac{1}{2} Re(i\omega \mathbf{B} \cdot \mathbf{H}^*), \tag{5}$$

where $\mathbf{J}$ is the electric current density, $\mathbf{E}$ is the electric field strength, $\mathbf{B}$ is the magnetic induction, and $\mathbf{H}$ is the magnetic field strength.

### 2.4. Boundary Conditions

Simulated system is supposed to be the periodic element of a matrix. For that reason, the periodic-type ports and periodic boundary conditions were used in order to model the repeating element of the surface (Figure 2a).

For electromagnetic boundary conditions we used two periodic ports: "on-type" and "off-type". The port is characterised by power, frequency, and electric field change formulation. The power was 30 mW. Electric mode field amplitude was changing in accordance with the following expression:

$$E_0 = e^{-i \cdot k_x \cdot x} \cdot e^{-i \cdot k_y \cdot y} \tag{6}$$

where $k_x$ and $k_y$ are wave vector components in $x$ and $y$ directions, $i$ is an imaginary unit.

Periodic boundary conditions were represented with the following equations:

$$\mathbf{E}_{dst} = \mathbf{E}_{src} e^{ik_F * (r_{dst} - r_{src})} \tag{7}$$

$$\mathbf{H}_{dst} = \mathbf{H}_{src} e^{ik_F * (r_{dst} - r_{src})} \tag{8}$$

where indexes "dst" and "src" mean source and destination boundaries, index "$F$" means floquet periodicity, $r$ is a radius-vector.

The excitation "on-type" port was described by the following equation:

$$S = \frac{\int_{\delta\Omega} (\mathbf{E} - \mathbf{E_1}) \cdot E_1}{\int_{\delta\Omega} \mathbf{E_1} \cdot \mathbf{E_1}}, \tag{9}$$

where $E_1$ and $E_2$ are electric fields on the port 1 and port 2, respectively.

The "off-type" port was described by the following equation:

$$S = \frac{\int_{\delta\Omega} \mathbf{E} \cdot \mathbf{E_2}}{\int_{\delta\Omega} \mathbf{E_2} \cdot \mathbf{E_2}}. \tag{10}$$

For thermal processes description, a constant temperature equalled to 293.15 K was set on the substrate lower surface. All the lateral vertical boundaries of the simulated system were described by means of periodic conditions:

$$-\mathbf{n}_{dst} \cdot \mathbf{q}_{dst} = \mathbf{n}_{src} \cdot \mathbf{q}_{src} \tag{11}$$

where $q_{src}$ and $q_{dst}$ are heat flux density on source and destination boundaries, and $n_{src}$ and $n_{dst}$ are normal vectors on source and destination boundaries.

## 3. Results

### 3.1. Filter

In the ideal case, filters are made of self-standing metal plates that work as good electric conductors and have high reflection; no substrates are used. In our case, the film is made of semimetal material that results in lower reflection amplitude (Figure 3). Film is deposited on the dielectric substrate which also affects the electromagnetic wave reflection. Moreover, substrate can produce its own resonances in the reflection amplitude spectrum (Figure 3b). These conditions complicate the optimisation process compared to metal filters. The change in period *P* helps to optimise the bandwidth and to decrease the side resonances (Figure 4). The change in cross width *W* and length *L* allows the adjustment of reflection amplitude and resonance position, almost with no impact on the side resonances (Figure 3a).

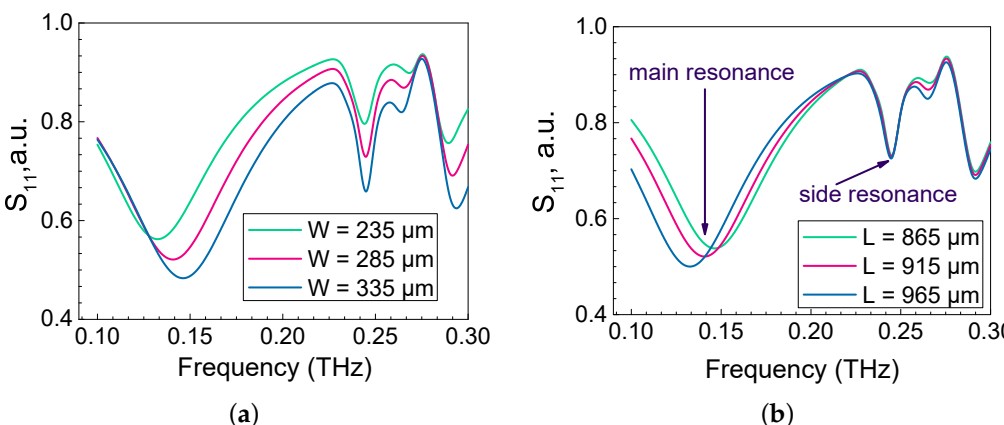

(**a**)                                        (**b**)

**Figure 3.** The example of reflection amplitude spectrum adjustment for $Bi_{88}Sb_{12}$ films on a polyimide substrate. (**a**) the impact of cross width (W) change (cross length = 915 µm and period = 1267 µm) (**b**) the impact of cross length (L) change (cross width = 285 µm and period = 1267 µm).

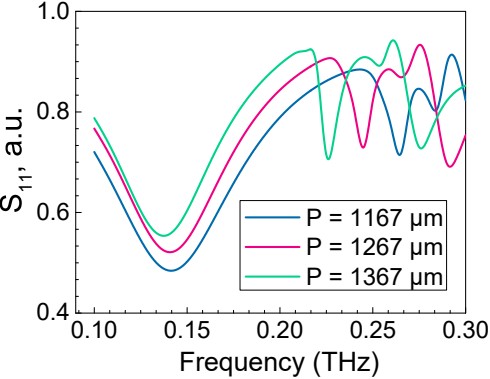

**Figure 4.** The example of reflection amplitude spectrum adjustment for $Bi_{88}Sb_{12}$ films on a polyimide substrate due to a period (P) change (cross length = 915 µm and width = 285 µm).

Changing the geometry of the system we found the parameters combinations allowing the obtaining of higher values of quality factor. The resulting geometric parameters of FSS are listed in Table 5.

**Table 5.** Optimized geometry parameters of FSS for 0.1 and 0.14 THz. P—period, L—length, W—width.

| Frequency, THz | Substrate | P, μm | L, μm | W, μm |
|---|---|---|---|---|
| 0.1 | mica | 1511 | 1358 | 380 |
| 0.1 | PM | 1511 | 1380 | 380 |
| 0.14 | mica | 1267 | 950 | 285 |
| 0.14 | PM | 1267 | 990 | 285 |

The reflection amplitude spectrum ($S_{11}$ parameter) corresponding to the system with geometry parameters from Table 5 are shown in Figure 5.

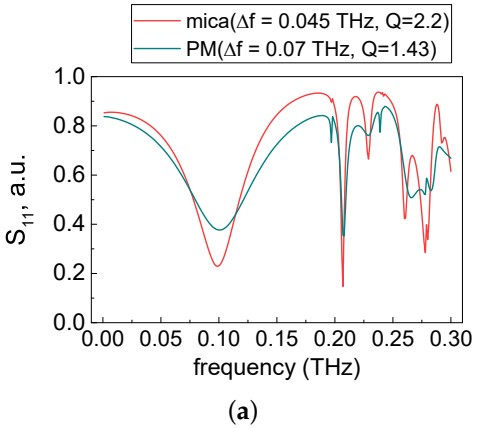 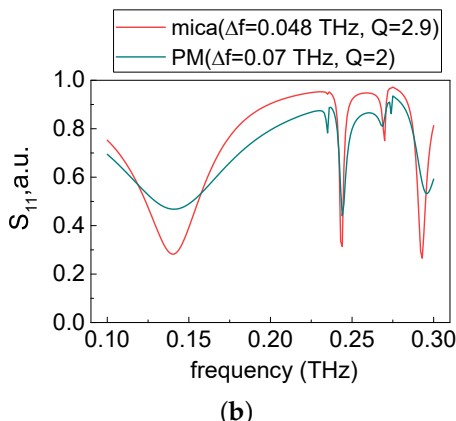

(**a**)          (**b**)

**Figure 5.** Reflection amplitude spectrum of $Bi_{88}Sb_{12}$-based FSS on different substrates under the exposure at different frequencies: (**a**) under 0.1 THz exposure (**b**) under 0.14 THz exposure.

A minimum of reflection can be seen at required frequencies. Q-factor, that depends on the resonance frequency, is higher for the 0.14 THz case: 2 and 2.9 for FSS on PM and mica substrates, respectively. Q-factor for 0.1 THz case equals to 1.43 and 2.2 for FSS on PM and mica substrates, respectively. Side resonances, that correspond to the substrate, are blueshifted by 0.1 THz relative to the main resonance. The impact of substrate is more significant in case of mica substrate usage when the reflection amplitude decreases and the transmission amplitude increases.

### 3.2. System Heating Due to THz Absorption

#### 3.2.1. Time Step Settings of the Solver

The THz wave period is around 10 ps. For a line of electromagnetic tasks solver step must be lower than the period. As in real conditions, times up to several seconds are under the interest it is required to perform the calculations up to the same times. However, heating process can be examined at time steps higher than the wave period. We made the comparison between three models: (1) with time step lower than the wave period (2 ps), (2) with time step comparable with the wave period (10 ps), (3) with time step higher than the wave period (50 ps). The results can be seen in Figure 6a. No significant change in the results has been traced. Hence, in the case of electromagnetic heating calculation, time steps higher than the wave period can be used.In case of longer exposure, higher time steps are required in order to reduce the calculation time. In Figure 6b temperature increase calculated with higher time steps (1, 100, and 250 μs) is shown. It can be seen that at 500 μs all the curves begin to coincide, and temperature increase function becomes the same for all solutions.

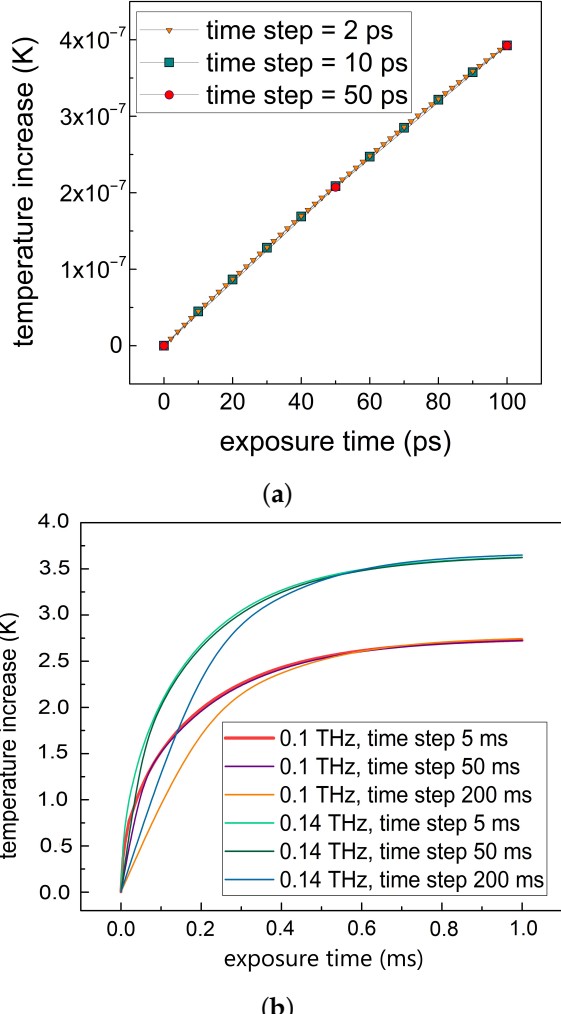

**Figure 6.** Temporal temperature increase in the film under the exposure calculated with different solver time steps. (**a**) the variations of small time steps at initial exposure times (example of 0.14 THz). (**b**) the variations of higher time steps at longer exposure times. The results are given for PM substrate with $\kappa$ = 1 W/(m· K).

### 3.2.2. The Impact of Thin Film Description on Temperature Increase Function

In finite elements approach, thin objects can be described in two general ways: (1) directly represented by a 3D object or (2) described with the analytical expressions.

We compared two model types: (1) 3D and (2) 3D with 2D description of the thin film. Time step was 100 ps. The 2D description has been performed using special options: "thin layer" in heat transfer interface and "transition boundary condition" in electromagnetic waves interface. The equations that are used for 2D description are listed in our previous publication [48]. The comparison between temperature change in 2D and 3D case showed that 2D description of film gives a temporal temperature function close to the linear (Figure 7a).The extrapolation of linear function gives unrealistic values of temperature. 3D model gives a function with the closest approximation in the form of natural logarithm. This is in good agreement with Newton's law of heating (and cooling) [49] that uses exponential growth function with the power containing time and thermophysical properties of the material.

A typical view of temperature field in the system obtained with the 3D model is given in Figure 7b.

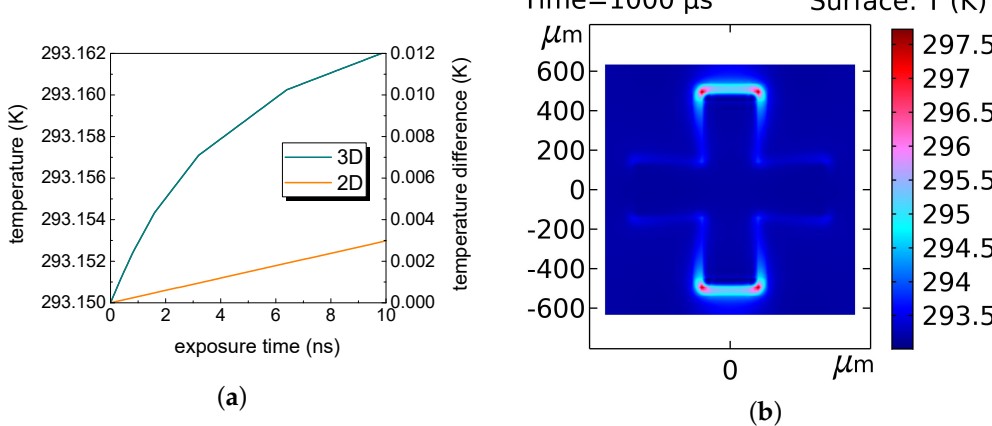

**Figure 7.** (**a**) Temporal temperature increase in the film obtained with 3D and 2D description. The example of filter on polyimide (PM) substrate under the 0.14 THz radiation exposure. (**b**) Typical temperature distribution in the film. The example of film on PM substrate under the 0.14 THz radiation exposure.

### 3.2.3. The Impact of FSS Geometry and Substrate on Thermal Response

Using the 200 µs solver time step we obtained the heating curves for films under the THz exposure. Resulting temperature in the film depends on two factors: (1) film optical properties, defining radiation absorption causing the heating, and thermal properties defining temperature and its distribution; (2) optical and thermal properties of the substrate. It is important to make a separated study of film and substrate properties impact on the processes. It can give deeper comprehension of substrate impact on film temperature.

Firstly, we made the comparison between substrates optical properties impact on the resulting film temperature. In this model case, it was necessary to exclude the impact of difference in thermal properties of the substrates. For this purpose, thermal conductivity coefficient was taken equal for mica and polyimide (both equal to 0.5 W·m$^{-1}$·K$^{-1}$). The results obtained are shown in Figure 8. In accordance with these calculations, mica substrate causes higher electromagnetic heating of the system.

The maximum amount of electromagnetic energy that can be absorbed and converted into the heat is mostly defined by film optical properties. The resulting temperature field depends on thermal properties of film and substrate. However, temperature field also depends on the cross-section area of the film and its geometry (the presence of narrow or thin regions, angles, etc.). In case of continuous film without cross-element cut from it, resulting maximum temperature is much lower due to the continuous film geometry (Figure 8b) than in case of FSS (Figure 8a).

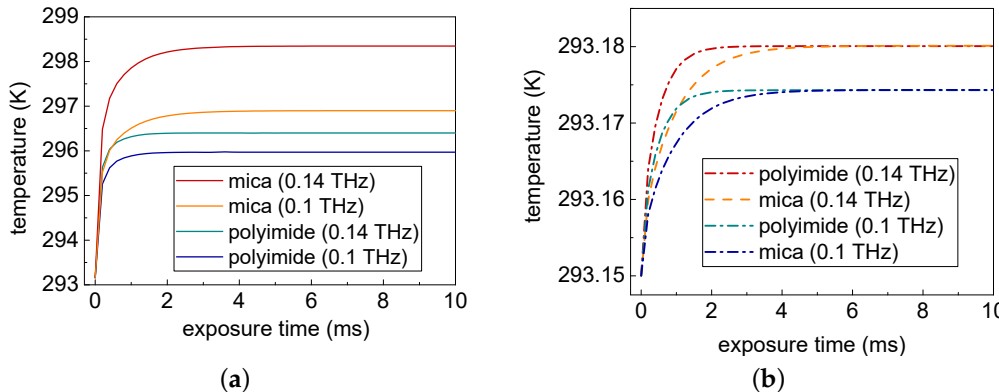

**Figure 8.** Temperature increase in the films on substrates with equal thermal conductivity ($0.5\ \text{W}\cdot\text{m}^{-1}\cdot\text{K}^{-1}$ in cross-plane and $5\text{W}\cdot\text{m}^{-1}\cdot\text{K}^{-1}$ in in-plane) but different dielectric properties. (**a**) FSS element. (**b**) continuous film.

Secondly, we studied the impact of substrate thermal conductivity on temperature in the film. In this model case, we took a substrate with PM optical properties and changed its thermal conductivity in the range from 0.3 to $5\ \text{W}\cdot\text{m}^{-1}\cdot\text{K}^{-1}$. These values of thermal conductivity are typical for mica and polyimide. Thermal conductivity of the substrate was assumed to be isotropic. Figure 9a shows that decrease of thermal conductivity of substrate results in the increase of thermal response in the film. This is due to the fact that smaller part of heat flux can penetrate into the substrate with low thermal conductivity. The dependence of film temperature on substrate thermal conductivity has a nonlinear behaviour that can be described by a power law (Figure 9b).

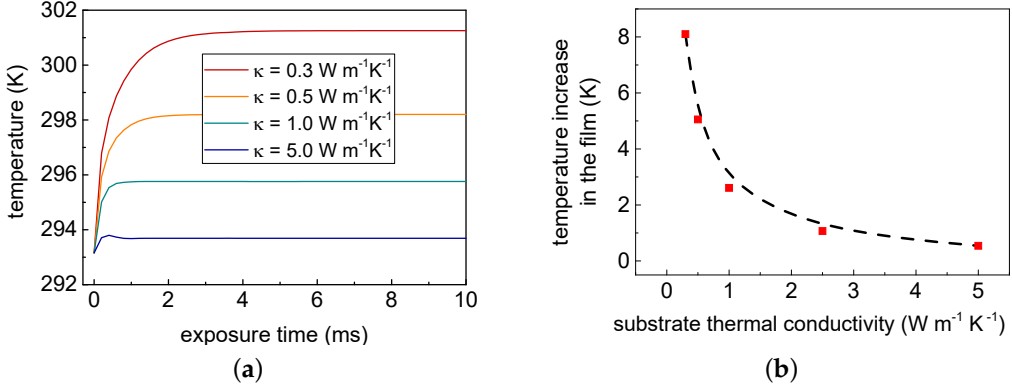

**Figure 9.** Temperature increase in the FSS on polyimide substrate under the 0.1 THz, obtained for different thermal conductivity of the substrate. (**a**) time dependence of temperature. (**b**) saturation temperature value at 10 ms as a function of thermal conductivity.

In the results we presented not average but maximum temperature in the film. This is due to the fact that for thermoelectric system not temperature itself but temperature gradient is essential. While the lateral surfaces of the simulated element had very low temperature changes, maximum temperature always corresponded to the right angles of the cross. Hence, significant temperature gradient directed from film periphery to the cross element can be expected.

## 4. Discussion

The reflection amplitude spectra show that thin films based on $Bi_{88}Sb_{12}$ solid solutions can be potentially used for frequency selective surfaces development for a THz frequency range (e.g., 0.1 and 0.14 THz). In the context of detection purpose, that means that the sensitivity of detector can be increased in 1.5–3 times in accordance with the quality factor

values. FSS on mica substrate gives higher Q-factor values in 1.5 times in comparison with FSS on polyimide substrate.

The absorption of THz radiation results in heating of the continuous $Bi_{88}Sb_{12}$ film up to the values around 10 mK. This value is not significant in the context of power generation device. However, for detecting purposes, the resulting temperature difference can be transformed into voltage signal and can be detected. With the Seebeck coefficient around 100 μV per 1 K of temperature difference the voltage of 1 μV can be expected from the element with the cross-section up to 1.5 mm$^2$.

The change in film geometry by cutting a cross-shaped element changes the process of radiation transmission. Such system works as the frequency selective surface increasing the transmission and decreasing the reflection. As a result, local heating of the system rises due to the increased absorption, and maximum temperature in FSS becomes several orders higher in comparison with continuous film. FSS geometry specifics (cross angles presence) results in nonuniform heat flux distribution and causes the temperature gradients increase near the cross angles (Figure 7b). Temperature gradient near the angles reaches the values up to 200 K/mm. Local temperature increase up to several degrees becomes possible on the element with the cross-section up to 1.5 mm$^2$. Such temperature difference response can correspond to the voltage of 100 μV that is promising for sensing applications.

Maximum possible temperature in the film (and, hence, temperature difference and temperature gradient) can be increased due to the use of substrate with low thermal conductivity coefficient, especially, lower than 1 W·m$^{-1}$·K$^{-1}$. The higher the thermal insulating properties of substrate the higher the temperature gradient in the film.

Time-dependent temperature function contains two regions: with fast temperature growth and high time derivatives and saturation region with stable temperature (see Figure 9a). On the one hand, transport coefficients (electrical and thermal conductivity) and Seebeck coefficient depend on temperature change. For detecting purposes first region may be of high interest due to possible change in material properties. On the other hand, saturation region may provide more stable temperature conditions and voltage drop.

Current work represents electromagnetic and thermal phenomena appearing in thermoelectrics under the THz radiation exposure. Additional phenomena can appear such as carriers generation, change in transport coefficients, and Seebeck coefficient at high frequencies. That potentially can influence the voltage generation and the effectiveness of energy conversion. These processes can be the promising directions of further research in this field.

## 5. Conclusions

For the first time, a thermoelectric frequency-selective surface has been proposed in the role of terahertz detector for the frequencies of 0.1 and 0.14 THz. The main advantage of detector is that FSS detecting surface represents an effective thermoelectric—$Bi_{88}Sb_{12}$ 150 nm thin film and can produce a voltage of the order of 0.1 mV from a 1.5 mm$^2$ surface due to the temperature difference of the order of several degrees. Apart from reasonable filtering quality factor of FSS (up to 3), its geometry results in a dramatic increase of temperature difference in the film from several hundredths of degree (in continuous films) to several degrees in FSS. That opens new prospects for thermoelectric conversion efficiency increase in the radiation detecting field as even insufficient heating due to electromagnetic energy absorption and resulting temperature gradients can be improved due to FSS geometry. Such detectors can be used in the sensing, imaging, nondestructive control, security systems, and medical diagnostics.

**Author Contributions:** Conceptualisation, M.K., N.K. and A.T.; methodology, A.T.; validation, A.Z., P.D. and D.Z.; formal analysis, A.A., A.N.; investigation, A.Z., P.D. and D.Z.; data curation, I.T.; writing—original draft preparation, A.T.; writing—review and editing, N.K., M.K.; visualisation, A.S.; supervision, M.K.; funding acquisition, M.K.; project administration, M.K. All authors have read and agreed to the published version of the manuscript.

**Funding:** This research was funded by Russian Science Foundation, grant number 19-72-10141.

**Institutional Review Board Statement:** Not applicable

**Informed Consent Statement:** Not applicable.

**Data Availability Statement:** Data are available from the authors on request.

**Conflicts of Interest:** The authors declare no conflict of interest.

## Abbreviations

The following abbreviations are used in this manuscript:

| | |
|---|---|
| THz | terahertz |
| FSS | frequency-selective surface |
| PM | polyimide |
| Q-factor, Q | quality factor |

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
