# Peer review of "FEM Simulation of Frequency-Selective Surface Based on Thermoelectric Bi-Sb Thin Films for THz Detection"

_photonics, doi:10.3390/photonics8040119_

Round 1

Reviewer 1 Report

The need for terahertz detectors with high sensitivity and a low noise floor is of increasing engineering interest. This is especially true for detectors that are capable of being operated at room temperature, as for the Bi-Sb thermoelectric detector studied by the authors. This simulations described in this manuscript are rigorously performed, and the conclusions are well presented and clearly described.

I am therefore pleased to recommend this manuscript for publication in Photonics. The manuscript contains a number of minor grammatical errors in its use of English, but these should and can be easily addressed at the journal sub-editing stage prior to publication.

Author Response

Thank you for the high evaluation of the work

Reviewer 2 Report

In the manuscript “FEM simulation of frequency-selective surface based on thermoelectric Bi-Sb thin films for THz detection” (photonics-1158666), Anastasiia Tukmakova et al. report the numerical results of basic electromagnetic and thermal properties of a periodic frequency-selective surface under the THz radiation. Compared with previously published works, such as Appl. Phys. Lett. 94, 181104 (2009), J. Opt. Soc. Am. B 27, 498-504 (2010), the present work employs a thermoelectric Bi-Sb thin film instead of conventional plasmonic metals. However, the shape of resonant unit cell and the filtering mechanism are all the same with those previous works. Therefore, the novelty and originality are quite limited.

In addition, the THz detection performance (as the title implies) is not examined completely: the authors stop at the temperature increase, but ignore the voltage production from thermo-electric effect.

My feeling is that this manuscript does not present a significant step-forward on the existing literature. Therefore I can not recommend it for publication in the emerging journal Photonics.

Author Response

Reviewer 2

Comment 1

Compared with previously published works, such as Appl. Phys. Lett. 94, 181104 (2009), J. Opt. Soc. Am. B 27, 498-504 (2010), the present work employs a thermoelectric Bi-Sb thin film instead of conventional plasmonic metals. However, the shape of resonant unit cell and the filtering mechanism are all the same with those previous works. Therefore, the novelty and originality are quite limited.

Answer

Firstly, in Appl. Phys. Lett. 94, 181104 (2009) the material under the study is HgCdTe. This material can perform rather good thermoelectric properties (thermal conductivity up to 2 W/m*K, Seebeck coefficient around 100 microV/K, but relatively low electrical conductivity around 2000 S/m) at low and room temperatures, but in this article, no attention has been paid to its thermoelectric nature. No attention has been paid to the temperature field formed in the system, which is a crucial parameter for thermoelectric effect efficiency prediction. Secondly, the concept of the work is HgCdTe as the basement for photoelectric effect, not photo-thermoelectric one. In addition, the configuration of the system under the study is different (rectangle on the substrate), and authors limit the calculations with 1D and 2D frequency-stationary calculations. In our work, the impact of 3D effects has been found and temperature evolution has been traced in time due to the frequency-transient calculations. In addition, this work studied the infrared frequency range, not THz frequencies. 

In second work (J. Opt. Soc. Am. B 27, 498-504 (2010)) the structure represented a bandpass filter (in our case, bandstop). The system was working in the transmission regime and operated in different frequency range: 3-7 THz (our case is 0.1-14 THz). And there were no thermoelectrics in the structure. 

Hence, the novelty of our work lies not so much in the filtering mechanism, as in the usage of thermoelectric material with high efficiency at room temperature. The main result shows a rather high temperature difference formed in the layer due to THz radiation absorption. This points to the perspective to develop a photo-thermoelectric detector with high efficiency, as thermoelectric effect is proportional to temperature difference.

Comment 2

In addition, the THz detection performance (as the title implies) is not examined completely: the authors stop at the temperature increase, but ignore the voltage production from thermo-electric effect.

Answer

In the discussion section we made the approximate estimation of voltage that can be formed under the calculated temperature difference and known Seebeck coefficient of the material (lines):

“The absorption of THz radiation results in heating of the continuous Bi88Sb12 film up to the values around 10 mK. This value is not significant in the context of power generation device. However, for THz radiation detection purposes, the resulting temperature difference can be transformed into voltage signal and can be detected. With the Seebeck coefficient around 100 µV∙K-1 of temperature difference the voltage of 1 ?V can be expected from the element with the cross-section up to 1.5 mm2.

The change in film geometry by cutting a cross-shaped element changes the process of radiation transmission. Such system works as the frequency selective surface increasing the transmission and decreasing the reflection. As a result, local heating of the system rises due to the increased absorption, and maximum temperature in FSS becomes several orders higher in comparison with continuous film. FSS geometry specifics (cross angles presence) results in non-uniform heat flux distribution and causes the temperature gradients increase near the cross angles. Temperature gradient near the angles reaches the values up to 200 K/mm. Local temperature increase up to several degrees becomes possible on the element with the cross-section up to 1.5 mm2. Such temperature difference response can correspond to the voltage of 100 ?V that is promising for sensing applications.”

The rough calculations presented in the discussion section give the orders of possible voltage generation: from 1 to 100 ?V. More accurate estimation of detector performance is the object of further experimental research.

Reviewer 3 Report

The paper reports the finite elements simulation of FSS based on Bi-Sb film on the top of a dielectric substrate. I would propose some major changes as follows:

  1. I am not sure if I grasp the novelty of the proposed analysis. Authors have to stress their novelty in comparison with other research.
  2. Authors have to justify the use Bi-Sb film. Moreover, it would be desirable to repeat the same analysis with other thin film options as well.
  3. Font size in all the Figures is too small. It should be increased.
  4. The titles, such as change.png in Figures 3, 4 should be removed.
  5. Figure is missing in Figure 6 (b). step 0.1 0.14. png appears instead.
  6. Authors are missing some recent works in thin films such as Tunable Plasmonic Properties and Absorption Enhancement in Terahertz Photoconductive Antenna Based on Optimized Plasmonic Nanostructures‎, etc.
  7. To increase the content of the manuscript, in my opinion, the Authors should fabricate a prototype of this structure in order to validate their numerical results.

Author Response

Reviewer 3

  1. Question

I am not sure if I grasp the novelty of the proposed analysis. Authors have to stress their novelty in comparison with other research.

Answer

As we could see from literature, there are some works combining metasurfaces and FSS with thermoelectric devices (e.g., absorption enhancers: Sharma, Nityanand, et al. "Metasurfaces for Enhancing Light Absorption in Thermoelectric Photodetectors." ACS Photonics 7.9 (2020): 2468-2473; L. Su, et.al. "Design and on-chip measurement of CMOS infrared frequency-selective-surface absorbers for thermoelectric energy harvesting," Asia-Pacific Microwave Conference 2011, 2011, pp. 461-464.). But, only few are about thermoelectrics as the very basis of FSS (Dong, Bowen, et al. "A thermally tunable THz metamaterial frequency-selective surface based on barium strontium titanate thin film." Journal of Physics D: Applied Physics 52.4 (2018): 045301. ). However, in this work FSS was considered as filtering surface, not a detector.

Hence, the novelty of the work is that for the first time, a thermoelectric frequency-selective surface has been proposed in the role of terahertz detector. And, secondly, calculations show an unobvious and new result - FSS helps to increase temperature gradient in the film that is significant for thermoelectric response and resulting voltage generation. These results are interesting for both photonics and thermoelectricity.

We stressed the novelty:

in Abstract (lines 9-14):

“We report for the first time about the simulation of THz detector based on thermoelectric Bi-Sb thin-filmed frequency-selective surface. We show that such structure can be both detector and frequency filter. Moreover, it was shown that FSS design increases not only a heating due to absorption, but a temperature gradient in Bi-Sb film by two orders of magnitude in comparison with continuous films. Local temperature gradients can reach the values of the order of 100 K/mm. That opens new perspectives for thin-filmed thermoelectric detectors efficiency increase.”

in the Conclusion (lines 306-316):

“For the first time, a thermoelectric frequency-selective surface based on room temperature thermoelectric Bi-Sb film has been proposed in the role of terahertz detector. The main advantage of detector is that FSS detecting surface represents an effective thermoelectric – Bi88Sb12 150 nm thin film, and can produce a voltage of the order of 0.1 mV from a 1.5 mm2 surface due to the temperature difference of the order of several degrees.

Apart from reasonable filtering quality factor of FSS (up to 3), its geometry results in a dramatic increase of temperature difference in the film from several hundredths of degree (in continuous films) to several degrees in FSS. That opens new prospects for thermoelectric conversion efficiency increase in the radiation detecting field as even insufficient heating due to electromagnetic energy absorption and resulting temperature gradients can be improved due to FSS geometry. ”

  1. Question

Authors have to justify the use Bi-Sb film. Moreover, it would be desirable to repeat the same analysis with other thin film options as well.

Answer

These materials have one of the highest thermoelectric properties at room temperature (high electrical conductivity in the range from 1*105 to 1*106 S/m, and Seebeck coefficient around up to 100 µV/K). Our previous research showed high permittivity, conductivity and subpicosecond relaxation time of carriers in Bi-Sb films. The composition with 12 % of Sb has the highest response to THz radiation turning on resulting in voltage drop around 0.1 mV.

We added this justification of Bi-Sb use and the corresponding references in the Introduction part in the lines 66-80 as the additional paragraph:

«These materials possess a narrow bandgap comparable with an energy of THz photons that can be used for practical detection applications. It was shown in {Mauser2017} that bismuth telluride and antimony telluride nanostructures can be a perspective basement for nanophotonic detectors with high responsivity and temporal response (337 µs). In {Huhn2013} a Bi/Bi-Sb-based thermoelectric THz antenna showed a temporal response of 22 µs without additional cooling. We showed in previously published works {Zaitsev2020 } that Bi-Sb thin films prepared by vacuum thermal deposition have perspective optical properties in the range from 0.2 to 0.8 THz: subpicosends relaxation time of carriers, high values of permittivity and conductivity. The composition with the antimony content of 12 percent (Bi88Sb12) seems to be the most promising. This composition has the highest response to THz radiation switching on: voltage drop of 0.1 mV along the Bi88Sb12 sample in comparison with 0.01 mV for Bi97Sb3 and Bi92Sb8 samples {Khodzitsky2021}. Temperature difference of several degrees in Bi88Sb12 film has been shown. One more advantage of the studied Bi-Sb films is that vacuum thermal deposition is not time and money consuming, and relatively easy method of films fabrication.»

  1. Question

Font size in all the Figures is too small. It should be increased. The titles, such as change.png in Figures 3, 4 should be removed. Figure is missing in Figure 6 (b). step 0.1 0.14. png appears instead.

Answer

The titles in Fig. 3,4 have been removed

Missing figures have been added

Font sizes have been replaced with the larger one.

  1. Question

Authors are missing some recent works in thin films such as Tunable Plasmonic Properties and Absorption Enhancement in Terahertz Photoconductive Antenna Based on Optimized Plasmonic Nanostructures‎, etc.

Answer

We added an additional paragraph to the Introduction part with thin and nano structures for detection based on thermoelectric (photothermoelectric) effect (lines 36-49):

“ Thermoelectric detectors can be a good alternative to bolometers, photoconductive antennas and Golay cells {Gric2018, Lewis2019}.

Thin films of graphene, black phosphorus, transition metal dichalcogenides, nitrides and carbonitrides are already widely studied as photothermoelectric detectors at room temperature {Wang2020}. Thermoelectrics can be used for detection from visible to infrared spectrum due to the possibility of precious temperature gradient control at the nanoscale volumes in resonant structures {Mauser2017}. Black-phosphorus {viti2016} and Se-doped black-phosphorus {viti2019} nano-transistors have been proposed as THz detectors working on thermoelectric response. Thermoelectric graphene-based detector has been presented in {viti2020} and in {liu2018} thermoelectric effect has been used to increase the responsivity of graphene-based THz detector. In {Szakmany2017} Au-based thermoelectric antenna has been fabricated for 0.6 THz frequency detection. To increase a temperature gradient in the system a hot junction width has been reduced to 100 nm. FSS and metasurfaces also have been studied as absorption enhancers for thermoelectric energy harvesters in infrared range {Sharma2020, Su2011}.”

  1. Question

To increase the content of the manuscript, in my opinion, the Authors should fabricate a prototype of this structure in order to validate their numerical results.

Answer

The experimental validation is on the preparation stage and will be performed later. Current work was planned as the preliminary stage for further experimental fabrication of frequency-selective surface. We added this to the Introduction part (lines 109-114):

“Prior numerical calculations are required for the simulation of a device with finite size and geometry, taking into the consideration electromagnetic and thermal phenomena, structure periodicity, boundary condition close to real operation conditions. Current work aims to perform a preliminary calculations for further thermoelectric FSS fabrication for THz radiation filtering and detection.”

Reviewer 4 Report

Review Report on photonics-1158666

The MS entitled ” FEM simulation of frequency-selective surface based on thermoelectric Bi-Sb thin films for THz detection” by Anastasiia S Tukmakova et al. reports on simulation of frequency-selectivity in the sub-THz range using the finite elements method. The paper addresses the ability of Bi-Sb to filter THz radiation based on its thermoelectric response (Seebeck effect).

The paper is well organized, and the English is relatively good.

The results in the paper are sound and are of interest for THz research and development.

I recommend the paper for publication.

Author Response

(The authors gave the same response as above.)

Round 2

Reviewer 2 Report

After carefully checking the response to reviewers' comments by the authors and the revised version of the submission, I think the authors have adequately addressed the comments. I recommend the manuscript acceptance in its current form. 

Reviewer 3 Report

Accept.